# Effects of Cage and Floor Rearing Systems on the Metabolic Components of the Uropygial Gland in Ducks

**DOI:** 10.3390/ani12020214

**Published:** 2022-01-17

**Authors:** Hehe Liu, Jiawen Qi, Qinglan Yang, Qian Tang, Jingjing Qi, Yanying Li, Jiwen Wang, Chunchun Han, Liang Li

**Affiliations:** Farm Animal Genetic Resources Exploration and Innovation Key Laboratory of Sichuan Province, College of Animal Science and Technology, Sichuan Agricultural University, Chengdu 613000, China; liuee1985@sicau.edu.cn (H.L.); miaowu777@126.com (J.Q.); yangqinglan@stu.sicau.edu.cn (Q.Y.); tq834817866@126.com (Q.T.); qjj15512296736@163.com (J.Q.); sicauliyanying@163.com (Y.L.); wangjiwen@sicau.edu.cn (J.W.); chunchunhai_510@163.com (C.H.)

**Keywords:** different rearing systems, feather appearance, metabolites, non-target metabolomics

## Abstract

**Simple Summary:**

With the development of the modern poultry industry, people have gradually transformed the floor rearing system into a cage rearing system. However, due to some factors, including the environment and management, the feather condition of caged ducks is generally poor, which impairs the healthy growth of ducks and the economic efficiency of breeders. It is believed that birds usually collect secretions from their uropygial gland and smear them on their feathers and cuticle scales during preening to improve their waterproofing and resistance to pathogens, thus protecting their health and growth. Therefore, we studied the uropygial glands of ducks in different rearing systems. The results showed that the cage rearing system affected the weight and metabolic components in the uropygial gland of ducks. Caged ducks have a lower relative weight of their uropygial gland and lower levels of certain amino acids and fatty acids that contribute to their development. This allows us to better understand the causes of the poor appearance of feathers in caged ducks.

**Abstract:**

Background: As a unique skin derivative of birds, the uropygial gland has a potential role in maintaining feather health and appearance. Cage-reared ducks usually have a worse feather condition than floor-reared ducks. We suspected that the metabolic components in the uropygial gland might play a vital role in their feather conditions. Methods: Herein, the uropygial glands of floor- and cage-reared ducks were weighed, and a nontargeted metabolic analysis was performed. Results: At 20 weeks of age, the relative weight of floor-reared duck uropygial glands was significantly higher than that of cage-reared ducks, indicating that the floor rearing system is better for inducing the development of uropygial glands. The nontargeted metabolic data revealed 1190 and 1149 differential metabolites under positive and negative ion modes, respectively. Among them, 49 differential metabolites were annotated between the two rearing systems. Three sulfur-containing amino acids, namely, 2-ketobutyric acid, L-aspartate-semialdehyde, and N-formyl-L-methionine, and some lipids, including inositol and sphingosine, might be responsible for the changes in plumage appearance among the various rearing conditions. Conclusions: The results of our study revealed the differences in the metabolic components of the uropygial gland in ducks reared under different rearing systems and found metabolic components to be possibly responsible for the poor feather condition of caged ducks.

## 1. Introduction

The uropygial gland is a unique skin derivative of birds located in the back of the tail base [1]. The uropygial gland forms and develops in the embryonic stage but degenerates in some adult birds [2]. Many waterfowl, i.e., ducks, petrels, pelicans, osprey, and oil birds, have well-developed uropygial glands [3,4]. Histological observations have shown that the gland opening is surrounded by two tufts of villi combined with the oily secretion. The uropygial gland comprises stratified epithelium containing secretory tubules filled with oil droplets. These oil droplets are discharged into the central cavity [5]. The uropygial gland structure and secretions’ composition are different in the various bird species, which ultimately leads to the gland having distinct functions in different species. The secretions of the uropygial gland are mainly composed of rich lipids, aliphatic monoesters, fatty acids (with various degrees of methyl branches), and long-chain monohydroxy waxy alcohols [6]. Using gas chromatography-mass spectrometry (GC-MS), researchers analyzed greenwood hoopoe uropygial gland secretions and reported their chemical composition, which includes aldehydes, aliphatic and heterocyclic aromatic amines, short-chain fatty acids, ketones, and dimethyl sulfides [7]. Moreover, researchers showed that the composition of the uropygial gland secretion may depend on the sex, age, and reproductive status of birds [8].

Past studies have illustrated the biological functions of the uropygial gland in mate selection [9], production performance [9], and reproductive performance [10]. After removing the uropygial gland, hens showed significantly lower mating activity than normal hens [9]. In addition, Giraudeau et al. installed a device on the uropygial gland of breeding ducks to prevent them from contracting their uropygial glands, and thus the secretions could not spread to their feathers. The results showed that the egg weight of the treatment group was significantly lower than that of the control group [11].

The uropygial gland may play an essential role in maintaining the integrity of feathers, since during preening, the birds collect uropygial gland secretions and smear them on their feathers; this is thought to improve waterproofing and increase resistance against pathogenic attacks [12,13]. Jacob et al. illustrated that uropygial gland secretions play potential roles in limiting bacterial contamination and maintaining good plumage integrity. The uropygial glands function as a defense mechanism to avoid the colonization of the feathers by pathogenic microorganisms and thus protect birds from infections and feather degradation [14,15].

With the development of the modern poultry industry, rearing systems for meat ducks have developed toward cage rearing systems and away from the traditional floor rearing mode. The plumages of the caged birds look worse than those raised in the conventional floor rearing system due to various factors, including management, environment, and genetics [16,17,18,19]. Plumage damage might increase the risk of abrasion and infection due to exposed areas of skin, while the loss of feather cover makes it hard to maintain a normal body temperature [20,21] and body condition [22] and can lead to mortality due to the cannibalism of denuded areas by other birds [23]. As a result, plumage damage might increase economic losses for producers due to increased flock mortality and feed consumption and decreased egg production [24,25,26]. Additionally, plumage damage also causes bird welfare issues.

Previously, Mi et al. found that providing an open water source for bathing can promote the development of the uropygial gland and preening behavior in ducks [27]. However, there is no report about the effect of the rearing systems on the metabolic content within the uropygial gland. Therefore, it was hypothesized that the metabolic components of the uropygial gland might also play a role in mediating plumage appearance changes under the impacts of rearing systems. Therefore, the present study was designed to investigate the changes in the uropygial gland in response to rearing in different production systems and to identify any differences in metabolites in the resultant uropygial glands. These primary data would help understand the mechanism linking the uropygial gland’s role in the plumage integrity of ducks to their rearing system.

## 2. Materials and Methods

### 2.1. Birds and Sampling

The Nonghua strain of ducks used in this experiment were all provided by the waterfowl breeding farm of Sichuan Agricultural University. The animal use protocol listed below was reviewed and approved by the Sichuan Agricultural University Animal Ethical and Welfare Committee. The study was conducted from August 2020 to January 2021. All ducks were raised using the floor rearing system until week 8. Then, 60 healthy female ducks with similar body weights were randomly divided into two groups. The first group of ducks was transferred to a cage rearing system, where the size of the stainless-steel mesh bed was 300 × 450 × 500 mm, with a mesh hole of 1.0 cm in diameter, set at a height of 50 cm from the floor. The other group of ducks continued to be reared in the floor rearing system with a 5 cm thick sawdust bedding covering a concrete floor. The stocking density was 0.87 m^2^ per duck. The temperature of the duck room was maintained between 20 and 30 °C, and the birds were fed a standard layer duck diet throughout the trial. At 16 and 20 weeks of age, 6 healthy ducks with similar body weights were randomly selected from each group. After exsanguination, a scalpel was used to cut the skin of the duck tail along the base of the villi, and the uropygial gland was removed. After removing the fat attached to the uropygial gland, it was weighed with a digital electronic balance (i2000, Yongkang, Zhejiang, China). Then, a portion of the left uropygial gland was taken, and the samples were placed in liquid nitrogen and quickly frozen. Then, the samples kept on dry ice were sent to Suzhou Panomick Biopharmaceutical Technology Co., Ltd. (Suzhou, Zhejiang, China), for metabonomic analysis.

### 2.2. Metabolite Extraction

Samples were thawed at 4 °C, and a 100 mg subsample was transferred into 2 mL centrifuge tubes. Then, 600 µL 2-chlorophenyl alanine (4 ppm) in methanol (−20 °C) was added and shaken for 30 s. After the addition of 100 mg glass beads, it was placed in a tissue grinder and ground for 90 s at 60 Hz. After ultrasonication at room temperature for 10 min and centrifugation at 4 °C for 10 min at 12,000 rpm, the supernatant was filtered through a 0.22 µm membrane to obtain the prepared samples for liquid chromatography-mass spectrometry (LC-MS). Twenty microliters from each sample was taken to generate quality control (QC) samples, and the rest was used for LC-MS analysis [28].

### 2.3. Chromatographic Conditions

Chromatographic separation was accomplished using a Thermo Ultimate 3000 system (UltiMate 3000, Thermo Fisher Scientific, Waltham, MA, USA) equipped with an ACQUITY UPLC® HSS T3 (150 × 2.1 mm, 1.8 µm, Waters, MA, USA) column maintained at 40 °C. The temperature of the autosampler was 8 °C. The mobile phase was 0.1% formic acid water-0.1% formic acid acetonitrile (positive ion mode) or 5 mM ammonium formate water-acetonitrile (negative ion mode). An injection of 2 μL of each sample was performed after equilibration. An increasing linear gradient of solvent B (*v*/*v*) was used as follows: 0–1 min, 2% formic acid acetonitrile/acetonitrile; 1–9 min, 2–50% formic acid acetonitrile/acetonitrile; 9–12 min, 50–98% formic acid acetonitrile/acetonitrile; 12–13.5 min, 98% formic acid acetonitrile/acetonitrile; 13.5–14 min, 98–2% formic acid acetonitrile/acetonitrile; 14–20 min, 2% formic acid acetonitrile (positive ion mode) or 14–17 min, 2% acetonitrile (negative ion mode) [29].

### 2.4. Mass Spectrometry Condition

Electrospray ionization mass spectrometry (ESI-MSN) experiments were conducted with a Thermo Q Exactive Plus mass spectrometer (Q Exactive HF-X, Thermo Fisher Technologies, Shanghai, China) with spray voltages of 3.5 kV and −2.5 kV in positive and negative modes. The sheath gas and auxiliary gas were set at 30 and 10 arbitrary units, respectively. The capillary temperature was 325 °C. The analyzer scanned over a mass range of m/z 81–1000 for a full scan at a mass resolution of 70,000. Data-dependent acquisition (DDA) MS/MS experiments were performed with a high-energy collision dissociation (HCD) scan. The normalized collision energy was 30 eV. Dynamic exclusion was implemented to remove some unnecessary information in the MS/MS spectra [29].

### 2.5. Qualitative and Quantitative Analysis of Metabolites

After the original data (xcms format) were converted into MZXML format using Proteowizard software (MacCoss, Mallick Research Labs and Insilicos, Seattle, WA, USA) [30], a series of operations, including peak identification, peak filtration, and peak alignment, were conducted using the XCMS package in R (v3.3.2). The main parameters were bw = 2, ppm = 15, peak width = c (5,30), mzwid = 0.015, mzdiff = 0.01, method = centWave. The obtained data matrix was exported to Microsoft Excel for subsequent analysis. Ion peaks with a coefficient of variation over 30% were deleted. To compare the data of different orders of magnitude, batch normalization of the intensity of the data was carried out.

### 2.6. Differential Metabolite Screening and Functional Analysis

The screening criteria for the differential metabolites were a *p* value ≤ 0.05 and a variable importance in projection (VIP) value ≥ 1. Metabolite identification first confirmed the precise molecular weight of the metabolite (molecular weight error < 30 ppm), and then the metabolites were annotated according to the MS/MS model in the Human Metabolome Database (HMDB) [31], Metlin [32], Massbank [33], lipid maps [34], MzCloud, and the self-built standard product Database of Panomics (Suzhou, China). Agglomerate hierarchical clustering was performed for each dataset using the PHEATMAP package in R (v3.3.2) [35]. Based on the MeTPA database [36], Kyoto Encyclopedia of Genes and Genomes (KEGG) enrichment was performed with differential metabolites to analyze the metabolic pathways related to the differential metabolites in each group and time point.

Orthogonal partial least squares discrimination analysis (OPLS-DA) [37] was performed on the metabolic data of all samples to assess the intra- and inter-group metabolite diversity. The data were normalized before analysis. Two screening criteria for differential metabolites were established: a fold change of 2 or 0.5 and a *p* value < 0.05 (Student’s *t*-test), and the VIP in OPLS-DA Model 1. In addition, visualization of the metabolites was performed using cluster analysis.

### 2.7. Statistical Analysis

The relative weight was the ratio of the absolute weight of the uropygial glands to the body weight. Weight data were analyzed using SPSS software, version 21 (International Business Machines Corporation, Armonk, NY, USA). An independent sample *t*-test was used for the analysis of group differences. The data are presented as the means ± SEM, and differences were considered to be statistically significant at *p* < 0.05.

## 3. Results

### 3.1. Changes in the Uropygial Glands’ Weight

Figure 1A,B shows an anatomical diagram of the uropygial glands dissected from one adult duck. We compared the absolute weight of the uropygial glands and their weight relative to the ducks’ bodyweights at 16 and 20 weeks of age (Figure 1). The weights of the ducks at 16 and 20 weeks are shown in Appendix A. We found that there was no significant difference in the total weight of the uropygial gland between the two groups (Figure 1C); however, the cage-reared ducks exhibited a lower relative weight of the uropygial gland than the floor-reared ducks (*p* < 0.05) at 20 weeks (Figure 1D), indicating that the cage rearing system is detrimental for the development of duck uropygial glands.

### 3.2. The Metabolic Components of Uropygial Glands Differed Significantly in Ducks from Different Rearing Systems

As shown in Figure 2, under the positive ion mode and negative ion mode, there were apparent differences in the metabolite components of the uropygial glands between floor-reared ducks and cage-reared ducks. In addition, most chromatographic peaks for cage-reared ducks were higher than those of floor-reared ducks. OPLS-DA analysis is extremely useful for predicting the differences between groups. As shown in the OPLS-DA score results (Figure 2), the metabolic components of the uropygial gland differed significantly between the ducks from the floor rearing system and the cage rearing system, indicating that the rearing systems had a significant influence on the metabolic components of the duck uropygial glands.

### 3.3. Identification and Classification of Metabolites in the Uropygial Glands

Metabolic component data for all 12 samples were obtained by analyzing the metabolic profiles. A total of 12,973 and 18,041 molecules were obtained in the positive and negative ion modes, respectively. The metabolites obtained were further annotated using HMDB, Metlin, Massbank, Lipid maps, Mzcloud, and the self-built standard product Database of Panomics. A total of 442 metabolites were annotated (Appendix A), including 102 carbohydrates and their derivatives, 75 amino acids and their derivatives, 23 exogenous substances, 70 unknown substances, 1 peptide, 49 nucleosides/nucleotides and analogs, 54 lipids and lipid molecules, 31 energy substances, 37 coenzyme factors, and vitamins. The relative content ratio of each variety of metabolites in the uropygial glands of cage-reared ducks and floor-reared ducks is shown in Figure 3.

### 3.4. Screening and Identification of Differential Metabolites

The difference in the metabolites was considered significant when the VIP value > 1 and the *p* value < 0.05 (from a *t*-test). A total of 1190 differential metabolites, including 908 upregulated metabolites and 282 downregulated metabolites, were screened under positive ion mode. In negative ion mode, 1482 differential metabolites were screened, including 1149 upregulated metabolites and 333 downregulated metabolites (Figure 4). A total of 49 differential metabolites were annotated (Appendix A, Figure 5); 42 molecules, such as myo-inositol, ketoleucine, and 2-ketobutyric acid, were downregulated, and seven, such as sphingosine 1-phosphate, normetanephrine, and bepridil, were upregulated in cage-reared ducks.

Metabolites with similar expression patterns may have similar functions or participate in regulating common molecular processes. A heatmap can help to cluster metabolites with similar expression patterns. The expression of the identified differential metabolites was divided into six main subclusters (Figure 5).

### 3.5. The Metabolic Pathway in the Gland Is Affected by Rearing Modes

A total of 32 KEGG metabolic pathways, such as ABC transport, glycine metabolism, and purine metabolism, were enriched based on the differential metabolites between the floor and cage rearing systems (Figure 6B, Appendix A). In addition, these metabolic pathways were enriched in five primary metabolic pathways, namely, metabolism, organismal systems, cellular processes, genetic information processing, and environmental information processing, and 13 secondary metabolic pathways, including amino acid metabolism, lipid metabolism, other amino acid metabolism, the sensory system, signal transduction, signal molecule and interaction, nucleotide metabolism, cofactor and vitamin metabolism, cell growth and death, membrane transport, energy metabolism, carbohydrate metabolism, and translation.

## 4. Discussion

In the modern poultry industry, during the overall transition of the rearing system, it was observed that the duck plumage looks worse in the cage rearing system than in the traditional floor rearing system. The uropygial glands play a vital role in maintaining the integrity of the plumage [38,39]. Our study observed that the relative weight of the uropygial glands of floor-reared ducks was significantly higher than that of cage-reared ducks, providing evidence showing the influence of the environment on uropygial gland weight, which might ultimately affect plumage integrity.

As an important part of systemic biology, metabolomics can identify changes in metabolites in different states by analyzing broad spectrum metabolites in organisms and clarifying the relationship between differential metabolites and their corresponding physiological and pathological conditions [40]. In this study, a detailed description of the metabolites of uropygial glands was provided through metabonomic analysis. A total of 12973 and 18041 molecules were found in the positive and negative ion modes, respectively, and 442 metabolites were finally annotated. Among them, lipids accounted for 12.245% of the total annotated metabolites, supporting previous findings that there are lipogenesis processes in chicken uropygial glands [41].

The differential metabolite analysis and metabolic pathway enrichment showed that the uropygial glands’ amino acids, lipids, carbohydrates, and energy metabolism were inhibited under cage rearing conditions. Amino acids are the precursors of various proteins synthesized by animals. Thus, they play an essential role in forming body structures, various metabolic processes, and stress resistance. In particular, sulfur-containing amino acids play a crucial role in plumage growth [42,43,44]. Our results showed that 10 amino acid metabolites were upregulated in the floor-reared ducks, which indicated that amino acid metabolism might be responsible for plumage appearance changes during the transition of the rearing system.

In addition, waterfowl can improve the waterproof function of their feathers by smearing oil on them [45]. In this study, a total of six different metabolites related to lipid metabolism were found, and five of them were upregulated in floor-reared ducks. They provided a metabolic basis showing that the uropygial glands play a greater role in feather waterproofing under the floor rearing mode. Inositol, tryptophan, and N-formyl-L-methionine were significantly increased in the floor-reared ducks. Inositol is an essential nutrient source for animals, and it may play a role in promoting hair growth, maintaining hair health, and metabolizing fat and cholesterol [46]. Research has shown that inositol deficiency in animals could lead to growth stagnation and physiological disorders [47,48]. At the same time, supplementation with inositol in the diet can reduce fearfulness in laying hens [49]. Van et al. found that feather pecking behavior is triggered by an acute reduction of 5-serotonergic turnover in the forebrain, as increasing 5-serotonergic turnover levels by increasing dietary tryptophan decreases feather pecking behavior [50]. N-Formyl-L-methionine is a derivative of methionine. The fourth primary wing feather length was increased with methionine supplementation [51]. This study may have found one of the reasons for the poor feather quality in cage-reared ducks compared to floor-reared ducks. Sphingosine is a type of sphingolipid that can promote cell survival and proliferation [52]. However, our experiment found that the level of sphingosine was higher in cage-reared ducks; therefore, the effect of sphingosine on feathers still needs further study. The mechanisms by which other differential metabolites affect feathers are not well understood and need further exploration.

## 5. Conclusions

In summary, we demonstrated that the cage rearing system could reduce the relative weight of duck uropygial glands. The nontargeted metabolic data illustrated 49 differential metabolites between the two rearing modes. Three sulfur-containing amino acids, namely, 2-ketobutyric acid, L-aspartate-semialdehyde, and N-formyl-L-methionine, and some lipids, including inositol and sphingosine, might be responsible for the changes in plumage appearance between the two rearing conditions. Therefore, the changes in the above metabolites in uropygial glands may be caused by the disturbance of the metabolic pathways, which may be partly responsible for the poor condition of feathers in caged ducks.

## Figures and Tables

**Figure 1 animals-12-00214-f001:**
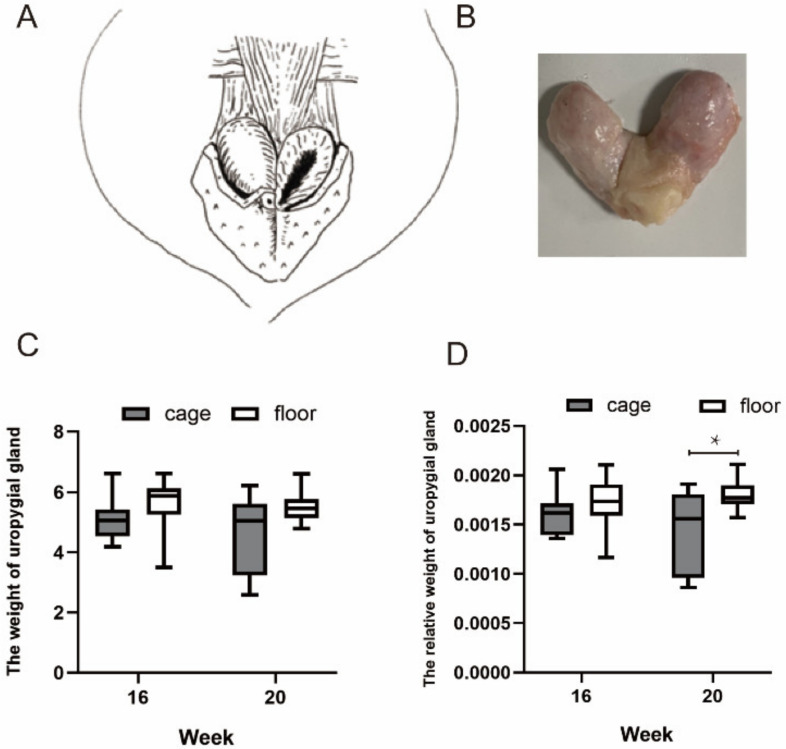
The uropygial glands of ducks. (**A**): Schematic diagram of the anatomical position of duck uropygial glands. (**B**): Image of uropygial glands dissected from a duck at 300 days old. (**C**): The absolute weight of the uropygial glands at 16 weeks and 20 weeks of age under different rearing systems. (**D**): The relative weight (absolute weight of the uropygial glands relative to body weight) of the uropygial glands at 16 weeks and 20 weeks of age under different rearing systems. The * indicate significant differences (*p* < 0.05).

**Figure 2 animals-12-00214-f002:**
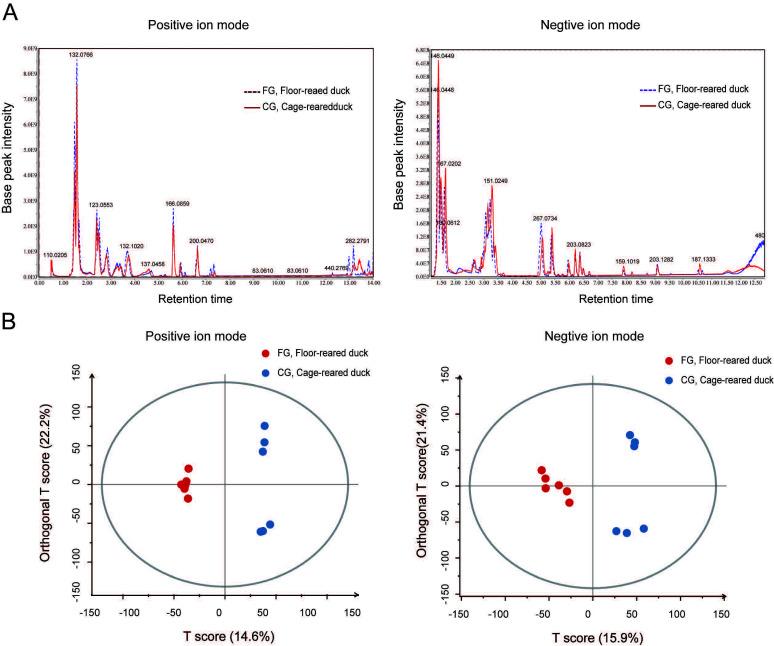
Separation of metabolic components of duck uropygial glands under different rearing systems. (**A**): The base peak chromatogram (BPC) of positive ion and negative ion modes. The BPC was generated with ion strength as the ordinate and time as the abscissa. The components separated by chromatography continuously enter the mass spectrometer, and the mass spectrometer constantly scans for data. Each scan obtains a mass spectrum. The strongest ion in each mass spectrum is selected for the continuous description. (**B**): OPLS-DA score of the metabolic components of duck uropygial glands for both positive and negative ion modes. FG, floor-reared duck; CG, cage-reared duck.

**Figure 3 animals-12-00214-f003:**
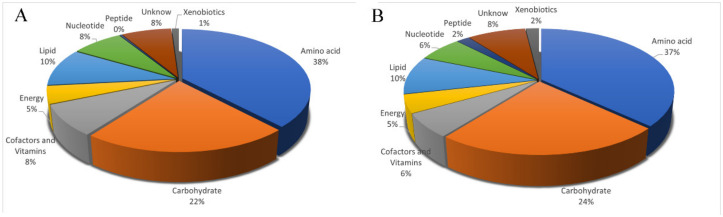
Pie chart of the relative content of various annotated metabolites in the uropygial gland of floor-reared ducks and cage-reared ducks. (**A**). The relative content of various annotated metabolites in the uropygial gland of floor-reared ducks. (**B**). The relative content of various annotated metabolites in the uropygial gland of cage-reared ducks.

**Figure 4 animals-12-00214-f004:**
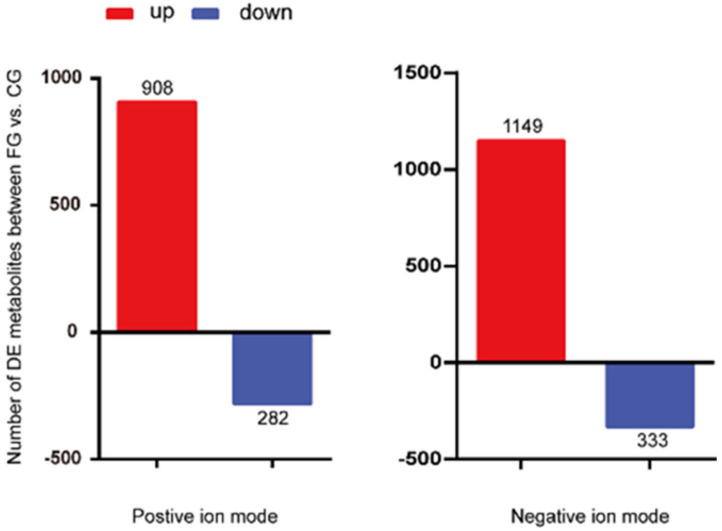
Differential metabolites between cage-reared ducks and floor-reared ducks under positive and negative ion modes. FG, floor-reared duck; CG, cage-reared duck.

**Figure 5 animals-12-00214-f005:**
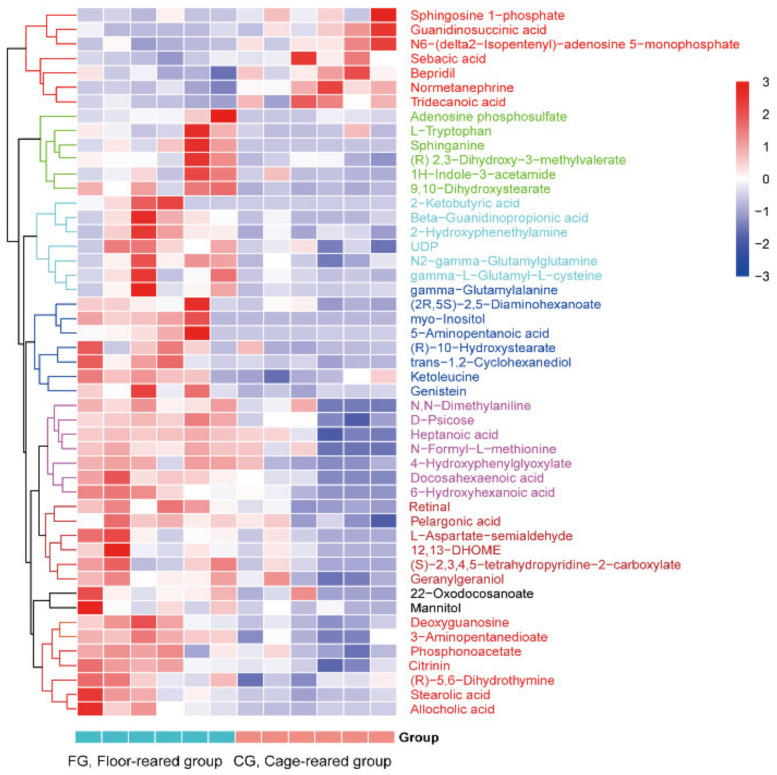
Cluster analysis shows differential metabolites between cage-reared ducks and floor-reared ducks. Each column represents a sample, and each row represents a metabolite. The color in the figure represents the relative content of metabolites in this sample. The closer the distance between two metabolite branches, the closer the expression level.

**Figure 6 animals-12-00214-f006:**
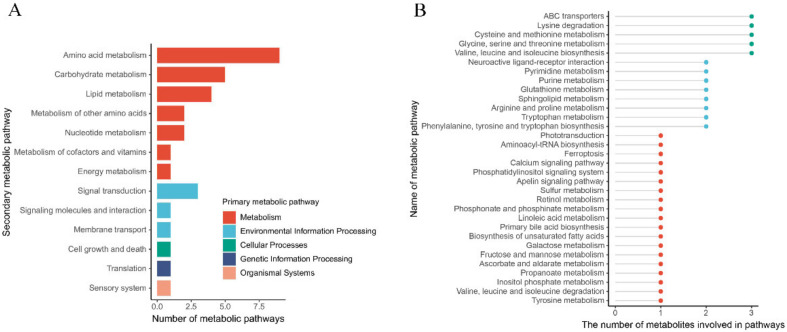
KEGG metabolic pathway enrichment was based on the differential metabolites between cage-reared ducks and floor-reared ducks in the uropygial glands. (**A**). primary metabolic pathways and secondary metabolic pathways. (**B**). Pathway analysis of the differential metabolites between cage-reared ducks and floor-reared ducks.

## Data Availability

The data presented in this study are available in Appendix A.

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
