# Peer review of "Effects of Cage and Floor Rearing Systems on the Metabolic Components of the Uropygial Gland in Ducks"

_animals, 2022, doi:10.3390/ani12020214_

Round 1

Reviewer 1 Report

Driven by profitability and efficiency, the cage rearing system is more and more common in modern duck production. Due to the change of habitat, increased stress less, limited space, and other reasons, cage reared ducks do not have as good plumage as free-range ducks. The manuscript provides a new perspective of explaining the poor feather quality in cage-reared ducks, which is valuable information in helping improve the management of cage-reared duck productions. The hypothesis that the uropygial gland might play a vital role in feather quality was well explained and investigated. Also, the metabonomic results help explain the mechanism of the change in the uropygial gland under different rearing systems. The study is well designed and the manuscript is relatively well prepared. However, there are quite some grammatical errors that need to be corrected before publication. Please see the following for specific and detailed comments.

L21 “Cage ducks have lighter uropygial gland” is not accurate in reflecting the result of this study, it should be “Cage ducks have a lower relative weight of uropygial gland”

L32 “Of them” should be “Among them”

L37-39 “metabolic components differences of uropygial ….. And found.. from the metabolic level” should be “differences in metabolic components of uropygial gland in ducks reared under different rearing systems, and found potential metabolic components responsible for the poor feather condition of caged ducks”

L59-62 “ After removing the uropygial gland……. than that of the uropygial gland absent” should be “After removing the uropygial gland of the hens, it was found that the number of mating activity with cocks in uropygial-gland-removed hens was significantly lower than the normal hens”

L62-65 “In contrast, after removing the olfactory bulb… and the uropygial gland ectomized group[15]” should be deleted. The content is not relevant to the current study, unless the authors want to further explain that it is through the olfactory system that cocks initiate their mating behaves. Additionally, this information was neither discussed anywhere else nor being used as an explanation for other measurements in this manuscript.

L70 Delete “It is believed that”

L80 “according to” should be “due to”

L81 “would” should be “might”

L82 “loss of feather cover results in difficulty maintaining body temperature [31, 34] and body condition” should be “loss of feather cover may make it hard to maintain normal body temperature [31, 34] and body condition”

L84 “would eventually” should be “might”

L88-90 “Therefore, we suspected the metabolic ….. under the impacts of rearing system transitions” should be moved to L93 before “Herein, the study”

L93-96 please rewrite “Herein, this study aims to illustrate …. uropygial gland on plumage integrity”. Suggestions are: “This study aims to investigate the potential changes of uropygial gland in responding to different rearing systems, illustrate the changes of metabolites in the resultant uropygial glands, to provide primary data for the study in the mechanism of the role of the uropygial gland on plumage integrity in ducks.”

L110-111 “6 ducks in each group were randomly selected and weighed”, why did not you weigh out all ducks used in the experiment? So that comparison of growth performance between the two groups can be made, this is also helpful information for the perception of uropygial gland data, which should be heavily affected by the bodyweight of ducks. Please provide the body weight information if they are available.

L113 Why do not you put the samples into liquid nitrogen for quickly frozen to avoid potential degradation of enzymes? It is understandable if acquiring liquid nitrogen is a challenge for you, but when it is available, samples should be quickly frozen as soon as possible after collection.      

L172 “Changes of the uropygial glands’ weight”, please provide the weight of ducks in the experiment if they are available to you.

L181 Figure 1 C and D, how come the legend of the figure is in a different order of the result bars. Please put the “grounds” in the front of “Cage” on top of the two figures as the legend.

L187-188 “The ducks in different groups were separated by the metabolic components of uropygial glands” should be “The metabolic components of uropygial glands differed significantly in ducks from different rearing systems”

L189 “As is shown in Figure 2 that under” should be “As shown in Figure 2, under…”

L190 “there are” should be “there were”

L193 “As seen from” should be “As shown in”

L194-195 “based on the metabolic …..been distinct significantly, indicated that” should be “the metabolic components of uropygial gland differed significantly between the ducks from the floor rearing system and cage rearing system, indicating that “

L207 3.3 identification and classification of metabolites in the uropygial glands, this section only provided an average content of metabolites of different categories, why can not you provide a comparison of both treatments? So that we may know whether or not the ratio differed in these ratios. It might also be very helpful information for future studies. Please provide if they are available to you

L212 delete “As a result”

L216 “was shown in Figure” should “is shown in Figure”

L217 delete “ the results showed that”, “Carbohydrates are the most” should be “Carbohydrates were the most”. Again, it will be great if the authors can provide a comparison of the groups in these ratios.

L 245 “Total” should be “A total of”

L245-253 please explain the differences of the five primary pathways and 13 secondary metabolic pathways between the two experimental groups.

L254-263 “ Of them….. amino acid metabolism” The paragraph was not drafted well enough to interpret the results of this section. Please rewrite the sentences. It shall be able to relate the metabolites to the changes of pathways they are related, and identify the pattern of the change. The word “following” in the context is confusing to readers.

L270-275 “In the modern poultry industry … traditional floor-reared system” is repetitive content, please delete the content.

L275 “the weight” should be “ the relative weight”  

L282-283 “ metabonomics provided a detailed description of the metabolites of uropygial glands” should be “a detailed description of the metabolites of uropygial glands was provided through metabonomic analysis.”  

L283 “Total” should be “A total of”

L285 “Of them” should be “Among them”

L298 “providing” should be “provided”

L300 as a part of the result, inositol and sphingosine should be described in the result section first before they were discussed in the discussion section.

L302 “sustaining” should be “maintaining”

L307 “main reasons”, I do not think the author can make this conclusion based on the measurements in this study. Although the difference in uropygial glands and their metabolites were detected, the authors can not exclude other factors like management stress, changing of habitat, limitation of movement, and other nutritional factors. Plenty of differences can be observed between the two rearing systems, for example, ducks may have access to insects, grass, and sands under the floor rearing system or free-range rearing system. It might be more reasonable to say that the study may have found one of the reasons that have caused the poor feather quality in cage rearing ducks in comparison to the floor rearing ducks.

Reviewer 2 Report

 In my opinion the topic of the article is very interesting and could provide relevant and novel information. However, I think the manuscript needs a thorough revision. More specific information needs to be added in the introduction, methodology and discussion. In addition, the results should be explained more clearly as well as the statistical methods used. I attach a word document with the specific information of the review. 

All the best.

Overall recommendation:

Review for “Effects of a cage and floor-reared system on the metabolic components of the uropygial gland in ducks” by Cristiano Liu et al.

In this experimental study analysed the uropygial gland of ducks in different rearing systems. They suggested that the weight and metabolic components of lipid glands in uropygial gland of ducks were affected by cage rearing system. However, since the data used is the weight of the gland, this could not be measured before and after, for this reason it had been better estimate de volume of uropigial gland with a digital calliper, which would have been the best method to determine the effect of the different rearing systems. In this way, I suggest that the best method to estimate the effect that different rearing systems have over the levels of certain amino acids and fatty acids in the uropygial gland is used the crossover experiment.

I found it to be a very interesting experimental study, however, the planification of the methodology it is not the best. There are important technical comments regarding the methodology and the statistic process. I think the results suggest needed improve and the discussion it is poor. I do some mayor comments and some suggestion.

Abstract:

Keywords:

Line 40: It should not include as keywords, words who are already in the title. Such as duck and uropygial gland. You should replace these words.  

Line 40: The keywords also should be in alphabetic order. Please change.   

Introduction:

General comments:

The introduction it is not bad, however in my opinion one of the most important aspect of this studies is the metabolic components of the uropigial secretion. The specific information about this aspect is poor. For example, the uropygial gland secretion is composed by both volatile and non-volatile fractions. These components could be varying among species and age classes and may differ between sexes and other reasons. In this introduction there are few information about it.   

Specific comments:

Line 50: This manuscript did not analyse the histological aspect of uropygial gland, The information in this line it is unnecessary.

Lines 55-56: I suggest move this line before the “As a special secretion…” (in line 51).

Lines 74-76: To justify this paragraph you could include same extra reference from non-experimental studies, such as (Magallanes et al 2016 of Parasite and vectors).   

Materials and methods:

General comments:

I think that there are some aspects that the authors would had have consider before experimental design, such as the importance od body condition, age, or, large wing areas. According to study of (Møller and Laursen 2019 Avian Research) “The Eider is a sea-duck that spends almost its entire life in sea water emphasizing water-proofing of the plumage. The size of the uropygial gland increased with body mass in males, but not in females, and it increased with age. The size of the uropygial gland decreased during winter. Eiders with small uropygial glands grew their feathers at a fast rate. Eiders with large wing areas had large uropygial glands”

Line 103-104: You said that each duck selected have similar body weight, but you did not weight these? The body condition it is an essential factor that it is positively relation with many aspects of uropygial gland. It would be very interesting to know if the ducks from one reared system or the other have gained or lost weight or have remained at the same weight.

Line 108: Is the stocking density equal in each reared system (0.87 m2 per duck)?

Between line 99-114: I miss important information such as what type of balance you use, have you considered the size of the animal to correct the weight of the gland, how did you carry out the dissection of the gland?

Line 116-145: I am not expert to Metabolite extraction, Mass spectrometry condition or Chromatographic conditions to use in analysed of uropygial secretion, I am surprised that there is no reference in any of these three sections. is it this the first time that those methods used for the analysis of uropygial secretion? If so, have any trials been done to test the efficacy, precision, accuracy, and repeatability of the methods used?  

Line 163: Please provide the meaning of all acronym such as KEGG.  

Line 155-170: In the statistical analyses you have carried out, have you considered the age of the individuals analysed or body condition? These are factors that have been shown in other articles to be related to the composition, properties, or volume of the uropygial grand.

Results:

Line 175-176: how did you calculate the p value? What factors did you consider?  What factors did you consider?

If you have not considered at least the age, the weight of the animals, ideally the body condition, it is necessary to repeat the analysis.

Line 185: Please include the information to estimate the relative weight of gland in the methodology.

Line 179-180: You suggest that “the floor-reared system is more beneficial for inducing the development of duck uropygial glands.” However, in figure 1 neither absolute weight nor relative weight increases between 16 and 20 weeks, so that floor rearing system only maintains the initial conditions, but the change from floor to cage causes a reduction in relative weight, (associated with the weight of the animal).  Apparent reduction in gland weight in the cage group between 16 and 20 weeks may be due to a reduction in the body condition of the animals. Where the reduction of the uropygial gland would be a direct consequence of the reduction of the body condition of the individuals.

Line 216-219: Why did you give one value of carbohydrates abundance? Did not you separate between groups the Identification and classification of metabolites in the uropygial glands? Are there differences in the annotated metabolites between rearing systems?

Line 244-263: I am not an expert I this topic, but it is to difficult extract de information of this paragraph, maybe this information could be included in a table summary.

Discussion:

General comment:

The discussion it is very short and need many improve in content and relation the result with previously studies.

Line 268-270: The uropygial glands are unique epidermal glands in birds. It is generally believed 268 the uropygial glands play a vital biological function in endowing the feather with water 269 resistance and increasing the feather's fat content”. I suggest delete this information; it is not necessary to include this in the discussion section.

Line 272-274: The role that uropygial secretion play in maintaining the integrity of plumage, it has been tested in several studies, such as (Møller and Laursen 2019; Shawkey et al 2003 or Ruiz-Rodriguez et al.2009…).

Line 275-276: The significative differences it is not in weight it is in relative weight.

Line 276-277: It is not the first evidence supporting the influence of environment on the uropygial glands' weight. (Møller and Laursen 2019) found that size of the uropygial gland decreased during winter. And (Magallanes et al 2020) found a positive relationship between uropygial gland volume and body condition, regardless of habitat (Urban and rural Environment) or (Magallanes et al 2020) showed that mean corrected uropygial gland size was on average 25% larger in bird species from tropical environment than in species from temperate environment.

Line 297-302: You said that found 6 different metabolites related to lipid metabolism were found, and 5 of them were up-regulated in floor-reared ducks. You discussed of them Inositol and Sphingosine, but you provide poor information and any few specific reference of this metabolite in birds. Please, improve more information about Inositol and talk about the rest of metabolites that were up-regulated in floor-reared ducks.

Conclusion:

This phrase “inositol and sphingosine, might be responsible for the changes of plumage appearance among various rearing conditions” it is a real conclusion, the rest of the text it is a summary of results. You need to explain what conclusion you could draw about your outcomes. Please rewrite the conclusion.

Figures

Figure 1: Maybe the boxplot could be mor appropriate.

What extra information does figure 4 provide compared to figure 2B?

Line 240: “Figure 5. Cluster analysis shows differential metabolites between cage-reared ducks and floor reared ducks.”  This information is it relative at 16 or 20 weeks?

Round 2

Reviewer 2 Report

I consider that the manuscript has improved but there is still some work to be done, I consider it very necessary that you provide more information on the statistical part. You can find the comments in the attached file. 

all the best,

reviewer 2
